# Routine Postoperative Antibiotic Prophylaxis Offers No Benefit after Hepatectomy—A Systematic Review and Meta-Analysis

**DOI:** 10.3390/antibiotics11050649

**Published:** 2022-05-12

**Authors:** Anastasia Murtha-Lemekhova, Juri Fuchs, Miriam Teroerde, Ute Chiriac, Rosa Klotz, Daniel Hornuss, Jan Larmann, Markus A. Weigand, Katrin Hoffmann

**Affiliations:** 1Department of General, Visceral and Transplantation Surgery, Heidelberg University Hospital, 69120 Heidelberg, Germany; juri.fuchs@med.uni-heidelberg.de (J.F.); miriamsarah.teroerde@med.uni-heidelberg.de (M.T.); rosa.klotz@med.uni-heidelberg.de (R.K.); katrin.hoffmann@med.uni-heidelberg.de (K.H.); 2Department of Pharmacy, Heidelberg University Hospital, 69120 Heidelberg, Germany; ute.chiriac@med.uni-heidelberg.de; 3Study Center of the German Surgical Society (SDGC), Heidelberg University Hospital, 69120 Heidelberg, Germany; 4Division of Infectious Diseases, Department of Medicine II, Freiburg University Medical Center, Faculty of Medicine, University of Freiburg, 79106 Freiburg, Germany; daniel.hornuss@uniklinik-freiburg.de; 5Department of Anesthesiology, Heidelberg University Hospital, 69120 Heidelberg, Germany; jan.larmann@med.uni-heidelberg.de (J.L.); markus.weigand@med.uni-heidelberg.de (M.A.W.)

**Keywords:** antibiotics, hepatectomy, liver surgery, antibiotic stewardship, meta-analysis, infection, surgical site infection, antibiotic resistance

## Abstract

Background: Prophylactic antibiotics are frequently administered after major abdominal surgery including hepatectomies aiming to prevent infective complications. Yet, excessive use of antibiotics increases resistance in bacteria. The aim of this systematic review and meta-analysis is to assess the efficacy of prophylactic antibiotics after hepatectomy (postoperative antibiotic prophylaxis, POA). Method: This systematic review and meta-analysis were completed according to the current PRISMA guidelines. The protocol has been registered prior to data extraction (PROSPERO registration Nr: CRD42021288510). MEDLINE, Web of Science and CENTRAL were searched for clinical reports on POA in hepatectomy restrictions. A random-effects model was used for synthesis. Methodological quality was assessed with RoB2 and ROBINS-I. GRADE was used for the quality of evidence assessment. Results: Nine comparative studies comprising 2987 patients were identified: six randomized controlled trials (RCTs) and three retrospectives. POA did not lead to a reduction in postoperative infective complications or have an effect on liver-specific complications—post-hepatectomy liver failure and biliary leaks. POA over four or more days was associated with increased rates of deep surgical site infections compared to short-term administration for up to two days (OR 1.54; 95% CI [1.17;2.03]; *p* = 0.03). Routine POA led to significantly higher MRSA incidence as a pathogen (*p* = 0.0073). Overall, the risk of bias in the studies was low and the quality of evidence moderate. Conclusion: Routine POA cannot be recommended after hepatectomy since it does not reduce postoperative infection or liver-specific complications but contributes to resistance in bacteria. Studies into individualized risk-adapted antibiotic prophylaxis strategies are needed to further optimize perioperative treatment in liver surgery.

## 1. Introduction

Surgical site infections (SSI) are the most prevalent complication after abdominal surgery, developing in up to 20% of patients [1]. In patients with SSI, mortality increases by over 12%, making strategies to reduce SSI a priority [1]. As the skin barrier is pierced during surgery, bacteria enter the abdominal cavity. In order to avoid clinically significant complications from this spread, standard hygiene measures must be adhered to [2]. In addition to decontamination and disinfection measures, perioperative antibiotic prophylaxis is given in major abdominal surgery to prevent postoperative infections. Ideally, this prophylaxis is given within 120 min before the incision [3]. Despite standards for hygiene, operations lasting longer than 3 h, those associated with high blood loss and blood transfusions, or requiring a long period of anesthesia are major risk factors for postoperative infections [2]. Moreover, the length of stay in the intensive care unit, as well as the total length of hospital stay, systemic comorbidities, previous antibiotic therapy and the degree of contamination of the operation are all risk factors for SSI [4].

Current evidence in colorectal surgery, interventions with inherent potential for site contamination, shows that single-dose prophylactic administration of antibiotics is non-inferior to prophylactic administration of three peri-operative doses [5]. Sporadic and unsubstantiated administration of antibiotics disrupts microbiota within the body and drives resistance in bacteria, which can lead to intractable infections [6]. This instigates questions whether abdominal surgeries with lower contamination potential require the redosing of perioperative antibiotics at all.

In addition to synthetic, excretory and metabolic functions, the liver is an immunologic organ [7]. After hepatectomy, the remnant must establish a new equilibrium in order to support all necessary functions. This is a major regeneration stimulus that transiently leads to a decline in function as commonly seen in the liver’s synthetic capabilities [8]. Post-hepatectomy infections are a major driver of morbidity and mortality as well as an increasing burden on healthcare costs [6]. Infective complications occur in up to 50% of patients after hepatectomy and up to 80% of patients with post-hepatectomy liver failure (PHLF) [8]. PHLF is a major contributor to mortality after liver surgery and preventing it remains the priority in research and clinical practice. While infection can cause PHLF, it can also be its consequence as well as the reason for fulminant progression. Thus, strategies to reduce postoperative infection in hepatectomy patients must be investigated to identify and select the most effective ones.

The aim of this systematic review and meta-analysis is to assess the efficacy of postoperative antibiotic prophylaxis in patients after hepatectomy.

## 2. Methods

The systematic review and meta-analysis were performed and reported in accordance with the current PRISMA guidelines [9]. The protocol of this meta-analysis has been registered prior to data extraction (PROSPERO registration Nr: CRD42021288510) [10]. Only studies that fulfilled the following PICO criteria were included in this systematic review:Population: patients undergoing liver resectionsIntervention: postoperative antibiotics beyond the first postoperative day (POD1) (POA)Comparison: no postoperative antibiotics beyond POD1 (control)Outcome: SSI (overall, superficial, and deep/organ), remote infections, sepsis, PHLF, bile leakage/bilioma, and microbiota changesStudy design: comparative studies

An analysis was also performed if studies compared two postoperative antibiotic prophylaxis strategies.

### 2.1. Literature Search

A systematic literature search was carried out according to the recent recommendations using databases MEDLINE via PubMed, Web of Science and CENTRAL without language or date restriction [11]. The aim of the search was to identify all comparative reports on postoperative antibiotic prophylaxis after hepatectomy. The search strategy included the terms “hepatectomy” and “antibiotic”, as well as their synonyms. The full strategy is provided as Appendix A. An additional hand search was performed through the references of included studies. The last search was performed on 30 September 2021.

### 2.2. Study Selection

The study methodology was restricted to comparative studies. The design of comparative studies was unrestricted and included randomized-controlled trials (RCTs), propensity score-matched trials, prospective and retrospective reports. Comments, editorials, meeting abstracts, correspondence and reviews were excluded. The screening of titles, abstracts, as well as of full texts was carried out by two independent reviewers (AML and JF) and subsequently compared. All disagreements were resolved through discussion and consensus with consultation with another reviewer (KH).

### 2.3. Data Extraction

Data extraction was performed by two independent reviewers (AML and JF) using a standardized form prepared prior to extraction and adjusted based on the first three data extractions. Following raw data were extracted: title of the publication, year, author, country, journal, study design, number of study groups, the total number of patients, patient characteristics, factors, indications for hepatectomy, if and type of underlying liver disease was present, type of hepatectomy, rates of complications—SSI (overall, superficial and deep), remote infections, sepsis, post-hepatectomy liver failure (PHLF), bile leakage/bilioma, microbiota changes. The reviewers also noted the sources of funding for the included studies.

### 2.4. Statistical Analysis

Meta-analyses were performed using R (Version 4.0.3) packages “metafor”, “meta”, and “ggplot2”. Summary statistics were performed using an independent *t*-test or χ^2^-test when applicable. Forest plots present effect estimates. A random-effects model was utilized for all outcomes. I^2^ statistics were used to evaluate heterogeneity with an I^2^ value below 25% indicating low, and over 75% indicating high heterogeneity. Odds ratios and 95% confidence intervals for dichotomous endpoints were pooled with the Mantel-Haenszel method.

### 2.5. Critical Appraisal of Included Studies

The risk of bias was assessed with Cochrane risk of bias tool 2 (RoB2) for RCTs and the ROBINS-I tool for non-randomized studies included. Certainty of evidence was assessed using GRADE. The assessments were performed by two independent reviewers (AML and JF) with consultation by a third reviewer on demand.

## 3. Results

After the exclusion of duplicates, 1876 records were screened for eligibility. After titles and abstract screening, 20 reports were assessed for eligibility based on full texts. Ultimately, nine reports were included in the qualitative and quantitative synthesis [12,13,14,15,16,17,18,19,20]. Figure 1 provides a detailed report of the study selection process.

Based on the results of the full-text screening, the following outcomes could be evaluated in the quantitative analysis: infective complications (overall surgical-site infections, superficial and deep SSIs, remote infections and sepsis) and liver-specific complications (PHLF and bile leakage). Table 1 provides details of included studies.

### 3.1. Critical Appraisal of Included Studies

Risk of bias assessment was performed for randomized controlled trials using the Cochrane Risk of Bias tool, Version 2 (Table 2). For retrospective comparative studies without randomization, the ROBINS-I tool was used (Table 3). Overall, included studies had a low risk of bias.

### 3.2. Patient Demographics

Cumulatively, 2987 patients were included in the meta-analysis. Most patients were male and had undergone an open hepatectomy for HCC. The summary demographics are provided in Table 4. Patients that did not receive post-operative antibiotics beyond POD1 (control) versus those patients that did (POA), did not vary significantly in age, gender, indications for surgery, type of surgery performed, or whether the surgery was performed through laparotomy or laparoscopically (Table 4).

### 3.3. Infective Complications

The main outcome of included studies and of this meta-analysis is surgical site infections. Seven studies [12,13,14,15,16,18,20] provided data for surgical site infection in patients receiving no postoperative antibiotics (control) compared to those given antibiotics until at least postoperative day 3 (POD3). Patients in the control group did not show higher rates of surgical site infections, whether deep, superficial, or overall, compared to those who received POA (Figure 2).

Rates of overall remote infections did not differ significantly either and sepsis complications were comparable between groups (Figure 3).

A subgroup analysis of matched studies, as well as RCTs only was performed. Neither subgroup analysis revealed any significant effect of POA (Forest plots for subgroup analyses are provided as Appendix A).

Two studies [17,19] analyzed two regimens of prolonged POA—until POD2 (POA2) or for at least 4 PODs (POA4). Although there was no significant difference in both groups regarding overall and superficial surgical site infections, deep surgical site infections occurred significantly more frequently in patients receiving longer regimens (Figure 4).

Remote infections rates were similar in both groups of prolonged POA (OR 0.77 95% CI [0.17; 3.47]; *p* = 0.82).

### 3.4. Liver-Specific Complications

Liver-specific complications—PHLF or postoperative bile leakage—were reported in six studies [13,14,15,16,18,20]. Neither PHLF nor bile leakage occurred more frequently in either group (Figure 5).

Subgroup analysis for PHLF did not reveal any significant difference (OR 0.70 95% CI [0.00; 119136]; *p* = 0.82). For this outcome results from matched and RCT groups were identical. Postoperative bile leakage also showed no significant differences between groups (matched group: OR 1.45 95% CI [0.55; 3.83]; *p* = 0.57; RCT group: OR 1.79 95% CI [0.44; 7.35]; *p* = 0.58).

For two prolonged antibiotic regimens (POA2 and POA4), only bile leakage was reported in both studies. No significant difference was shown (OR 0.85 95% CI [0.00; 1887.02]; *p* = 0.83).

### 3.5. Pathogenic Isolates

The most common pathogenic isolates for control versus POA were provided by six studies comparing POA versus control [12,14,15,16,18,20] and are depicted in Figure 6.

Methicillin-resistant *Staphylococcus aureus* (MRSA) was detected significantly more as the pathogenic isolate in patients with routine POA, while *S. aureus* (without resistance) was detected significantly more often in the control group (*p* = 0.007293 and *p* = 0.000399, respectively, as determined by the χ^2^-test). The incidence of *S. aureus* as the pathogenic isolate regardless of the resistance was similar in both groups (*p* = 0.507111). All other pathogens did not show a statistically significant variation in this analysis. The association between deep SSIs and MRSA could not be tested due to insufficient raw data provided by individual studies.

### 3.6. Certainty of Evidence

GRADE (grading of recommendations assessment, development and evaluation) approach was used to rate the certainty of evidence. An overview of the main outcomes is provided in Table 5.

Overall, the quality of evidence was rated as moderate with two outcomes (PHLF and sepsis) downgraded due to the low number of studies.

## 4. Discussion

Liver surgery is a rapidly growing surgical specialty with expanding indications and novel techniques appearing and constantly evolving [21,22]. In addition to technique optimization, it is important to improve perioperative management. With complication rates reaching 31% for hepatectomy, validated strategies to reduce morbidity are wanted [23]. Postoperative antibiotic prophylaxis has been a subject of debate recently as more and more studies failed to show the benefit of this practice even in cases of intraoperative contamination [24,25,26]. There is an exceeding need for improving strategies for antibiotic use within clinical settings. Previous network meta-analysis of five studies has compared various antibiotic strategies and did not reveal a benefit of additional antibiotics, findings which were confirmed in the current meta-analysis of nine studies [27]. The growth and spread of multidrug-resistant bacteria lead to increased mortality, morbidity and healthcare cost [6]. A major driver of multidrug resistance is sporadic and unnecessary antibiotic use. An evidence-based approach to antibiotic administration is needed and with this, common, empirically-driven practices must be evaluated and verified. Current meta-analysis suggests an increase of resistance in pathogens after postoperative antibiotic prophylaxis without the benefit of a reduced rate of overall infections.

After hepatectomy, liver function suffers from a transient decline, which is predominantly evident by systematic measurements of protein synthesis and coagulation parameters [8]. How hepatic immunologic function is affected is less clear. Since antibiotics are predominantly metabolized by the hepatocytes, strategies to reduce potential injury to the liver are invaluable. During the perilous phase where the liver must regenerate to accommodate the demands of the organism, the question emerges if antibiotics provide additional support or a hurdle.

This meta-analysis provides first glimpses into the efficacy of postoperative prophylactic antibiotics after hepatectomy. Prophylactic use of postoperative antibiotics fails to show any significant improvement in terms of infective or liver-specific complications. In fact, prolonged administration of antibiotics for more than four days was associated with more deep surgical site infections. Although counterintuitive at first glance, antibiotics only select bacteria, which leads to increased resistance [6]. With more resistance, difficult-to-treat bacteria can spread and become problematic for the organism to clear. With drug-resistant bacteria becoming a major burden on healthcare due to high associated mortality, morbidity and treatment cost, reduction of resistance becomes an increasingly important research area [28]. Postoperative antibiotic prophylaxis after hepatectomy leads to increased multi-drug resistance in *Staphylococcus aureus*. Whether other resistances increase as well, e.g., in bacteria that were not primarily pathogenic, remains unclear. Long-term consequences of increased resistant bacteria, in particular MRSA, in patients after hepatectomy, too, remain unclear but can only be anticipated as undesirable.

Increasing evidence suggests that when it comes to antibiotics, less is more unless it’s sepsis. With this thought in mind, clinicians must critically reexamine antibiotic strategies. The appraisal for postoperative antibiotic prophylaxis after hepatectomy reveals the strategy as inadequate and superfluous, thus one that should be abandoned.

### Certainty of Evidence

The main limitation of the analysis is the limited number of studies on the topic. Simultaneously, with most RCTs, the quality of evidence is satisfactory. Various studies had different regimens implemented with most comparing no post-operative antibiotics versus three days. However, three doses versus nine, as well as zero versus seven days of antibiotics were also compared. This brings a certain heterogeneity into the analysis. Different antibiotics were utilized as prophylactics with principally flomoxef sodium administered.

The risk of bias in included studies was generally low and the certainty of the evidence was overall moderate based on the methodology of the studies. This is the highest certainty of evidence available to date.

## 5. Conclusions

Routine postoperative antibiotic prophylaxis cannot be recommended for patients after hepatectomy and the administration should be discouraged. Prolonged exposure to antibiotic prophylaxis leads to an increase in resistant bacteria and thus bears an additional risk for patients. Prophylactic antibiotic strategies must be assessed for their efficacy and safety to further optimize liver surgery. Studies into individualized risk-adapted antibiotic prophylaxis strategies are needed to further optimize perioperative treatment in liver surgery.

## Figures and Tables

**Figure 1 antibiotics-11-00649-f001:**
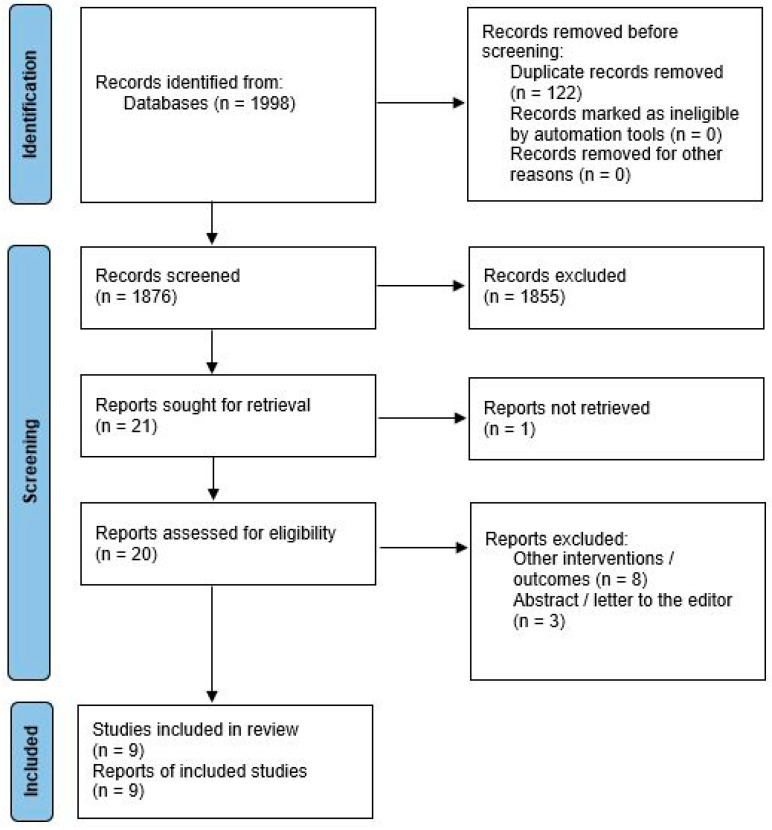
Study selection process.

**Figure 2 antibiotics-11-00649-f002:**
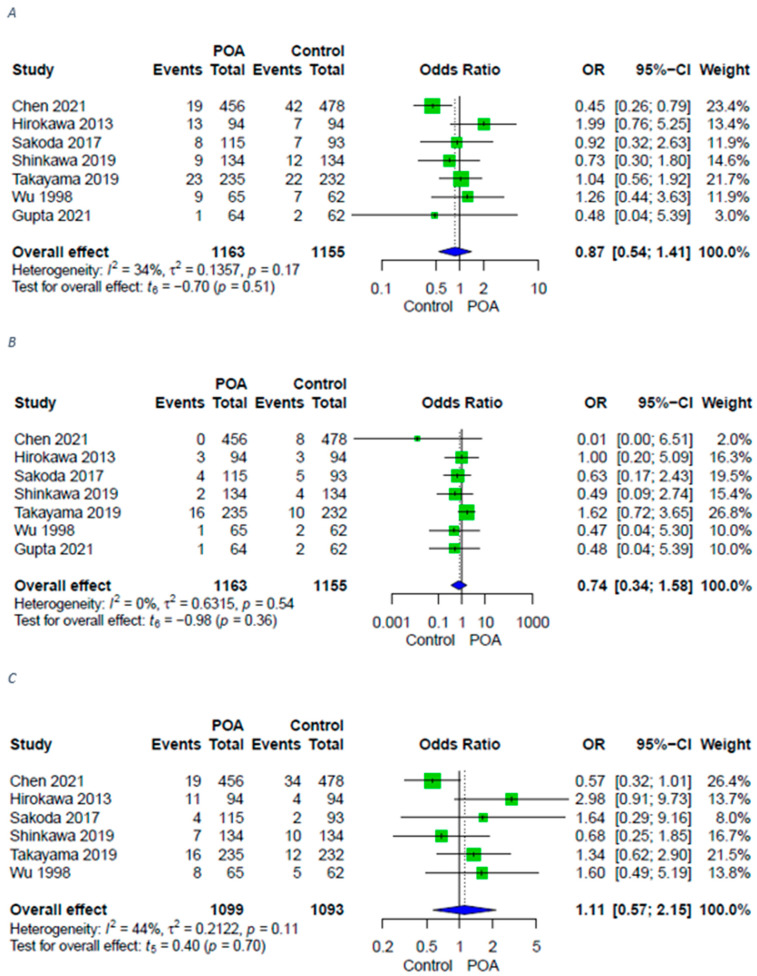
Forest plots for surgical site infections in patients with POA versus control. (**A**) Forest plot for surgical site infections in patients with POA versus control; (**B**) Forest plot for superficial surgical site infections in patients with POA versus control; (**C**) Forest plot for deep surgical site infections in patients with POA versus control. A random-effects model was utilized for all outcomes due to heterogenic methodological and clinical framework of included studies. Statistical heterogeneity was evaluated using the I^2^ statistics. An I^2^ value below 25% indicated low heterogeneity, while over 75% was considered high.

**Figure 3 antibiotics-11-00649-f003:**
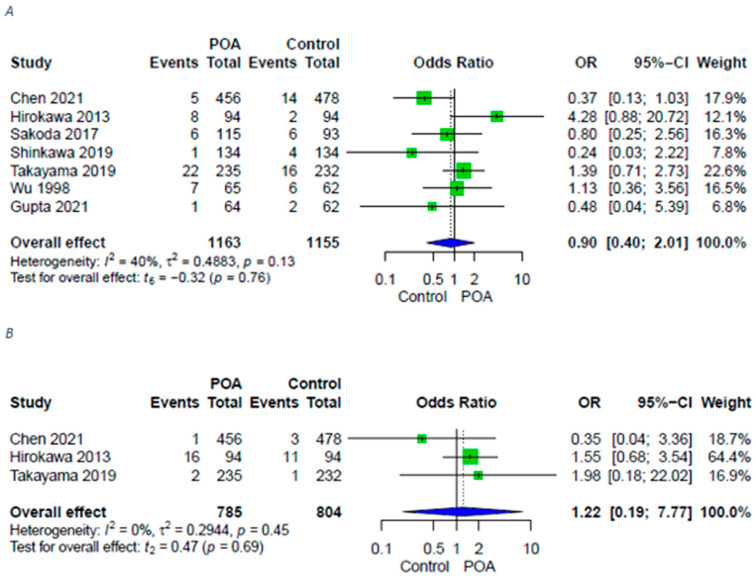
Forest plots for remote infections in patients with POA versus control. (**A**) Forest plot for remote infections in patients with POA versus control; (**B**) Forest plot for sepsis in patients with POA versus control. A random-effects model was utilized for all outcomes due to heterogenic methodological and clinical framework of included studies. Statistical heterogeneity was evaluated using the I^2^ statistics. An I^2^ value below 25% indicated low heterogeneity, while over 75% was considered high.

**Figure 4 antibiotics-11-00649-f004:**
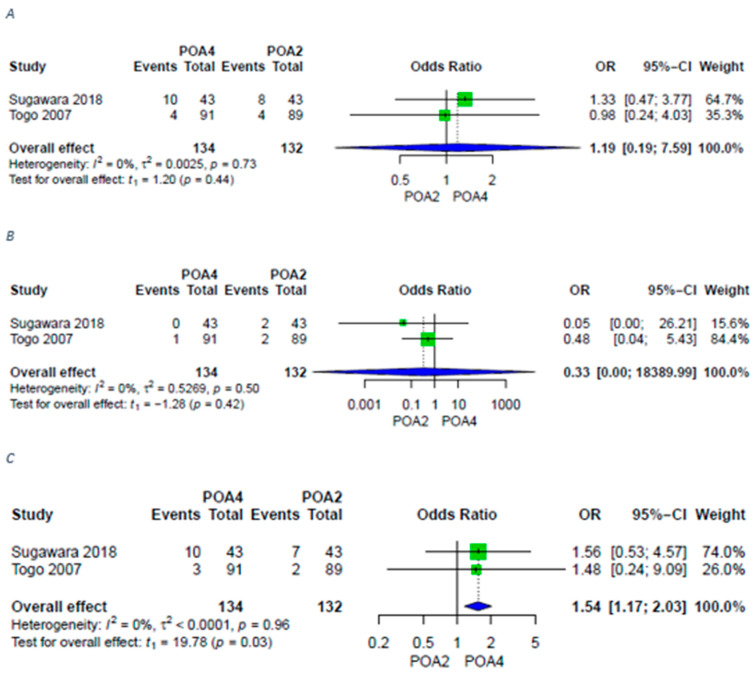
Forest plots for surgical site infections in patients with POA2 versus POA4. (**A**) Forest plot for surgical site infections in patients with POA2 versus POA4; (**B**) Forest plot for superficial surgical site infections in patients with POA2 versus POA4; (**C**) Forest plot for deep surgical site infections in patients with POA2 versus POA4. A random-effects model was utilized for all outcomes due to heterogenic methodological and clinical framework of included studies. Statistical heterogeneity was evaluated using the I^2^ statistics. An I^2^ value below 25% indicated low heterogeneity, while over 75% was considered high.

**Figure 5 antibiotics-11-00649-f005:**
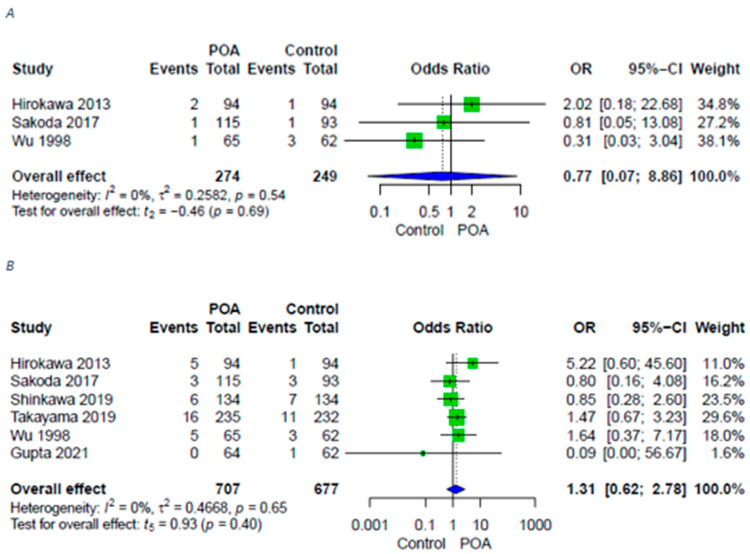
Forest plots for liver specific complications in patients with POA versus control. (**A**) Forest plot for PHLF in patients with POA versus control; (**B**) Forest plot for bile leakage in patients with POA versus control. A random-effects model was utilized for all outcomes due to heterogenic methodological and clinical framework of included studies. Statistical heterogeneity was evaluated using the I^2^ statistics. An I^2^ value below 25% indicated low heterogeneity, while over 75% was considered high.

**Figure 6 antibiotics-11-00649-f006:**
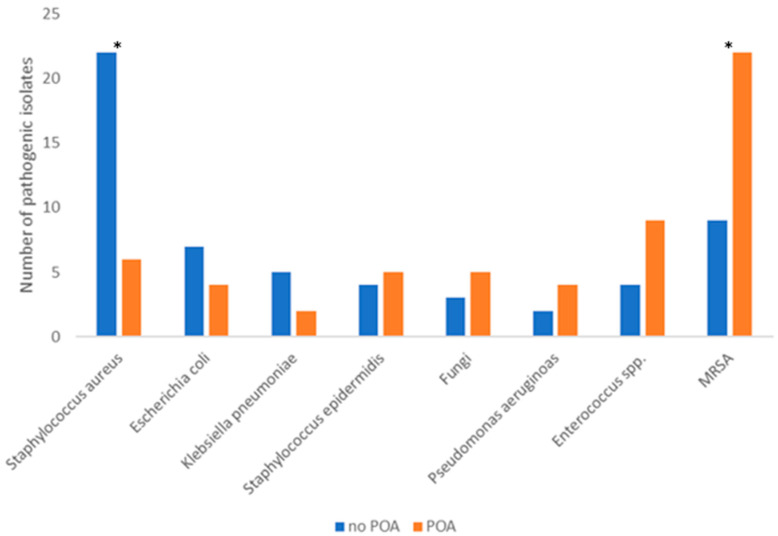
Most common pathogenic isolates. * statistically significant difference as determined by the χ^2^-test.

**Table 1 antibiotics-11-00649-t001:** Included studies.

Report	Study Design	Indications for Hepatectomy	Type of Hepatectomy	Method of Access	Number of Patients in Intervention Group	Number of Patients in Control Group	Duration of Intervention Regimen	Duration of Control Regimen	Antibiotic Investigated
Chen 2021	Retrospective	HCC	Unspecified	Unspecified	456	478	Various	None	Cephalosporins
Hirokawa 2013	RCT	Various	Major/Minor	Unspecified	94	94	3 days	None	Flomoxef sodium
Sakoda 2017	Retrospective	Various	Major/Minor	Open/Laparoscopic	115	93	3 days	None	Cefotiam
Shinkawa 2019	RetrospectiveSubgroups propensity score matched	Various	Major/Minor	Open/Laparoscopic	75	173	3 days	None	Flomoxef sodium
Sugawara 2018	RCT	Various	Major	Unspecified	43	43	4 days	2 days	Various
Takayama 2019	RCT	HCC	Major/Minor	Open	235	232	3 days	None	Flomoxef sodium
Togo 2007	RCT	Various	Major/Minor	Unspecified	91	89	5 days	2 days	Flomoxef sodium
Wu 1998	RCT	Various	Major/Minor	Unspecified	65	62	7 days	None	Cephazolin/gentamicin
Gupta 2021	RCT	Live liver donors	Major/Minor	Open/Laparoscopic	64	62	9 doses	3 doses	Piperacillin/Tazobactam

HCC: hepatocellular carcinoma; RCT: randomized controlled trial.

**Table 2 antibiotics-11-00649-t002:** RoB2 for included RCTs.

	Randomization Process	Deviations from Intended Interventions	Missing Outcome Data	Measurement of the Outcome	Selection of the Reported Results	Overall
Hirokawa 2013 [9]		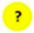				
Sugawara 2018 [14]						
Takayama 2019 [12]		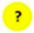				
Togo 2007 [15]		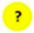				
Wu 1998 [13]		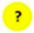				
Gupta 2021 [8]		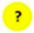				


 Low risk 
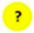
 Some concern 
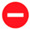
 High risk.

**Table 3 antibiotics-11-00649-t003:** ROBINS-I for included non-randomized comparative studies.

	Bias Due to Confounding	Bias in Selection of Participants into the Study	Bias in Classification of Interventions	Bias Due to Deviations from Intended Interventions	Bias Due to Missing Data	Bias in Measurement of Outcomes	Bias in Selection of the Reported Results	Overall
Chen 2021 [7]								
Sakoda 2017 [10]								
Shinkawa 2019 [11]								

Low, Moderate, High.

**Table 4 antibiotics-11-00649-t004:** Summary demographics.

	No AB	POAs	Level of Significance **
Age *	63.4 ± 11.9	62.5 ± 13.4	0.519
Gender			0.130
- Male	580	485	
- Female	228	226	
Indication			0.726
- HCC	782	809	
- CC	3	5	
- CRLM	48	40	
- LDLT	63	68	
- Other	10	14	
Type of surgery			0.215
- Major	231	226	
- Minor	565	481	
Mode of surgery			0.274
- Open	512	454	
- Laparoscopic	130	98	

HCC: hepatocellular carcinoma; CC: cholangiocarcinoma; CRLM: colorectal liver metastasis; LDLT: living donor liver transplantation; * Expressed as mean with standard deviation; ** Significance determined with an independent *t*-test for continuous and χ^2^-test for categorical variables.

**Table 5 antibiotics-11-00649-t005:** Certainty of the evidence for outcomes of the main analysis.

Outcome	№ of Included Studies	Certainty of the Evidence (GRADE)	Relative Effect(95% CI)
Surgical site infections	7 (4 RCTs, 3 retrospective)	⨁⨁⨁◯ Moderate	OR 0.87 [0.54; 1.41]
Superficial surgical site infections	7 (4 RCTs, 3 retrospective)	⨁⨁⨁◯ Moderate	OR 0.74 [0.34; 1.58]
Deep surgical site infections	6 (3 RCTs, 3 retrospective)	⨁⨁⨁◯ Moderate	OR 1.11 [0.57; 2.15]
Remote infections	7 (4 RCTs, 3 retrospective)	⨁⨁⨁◯ Moderate	OR 0.90 [0.40; 2.01]
Sepsis	3 (2 RCTs, 1 retrospective)	⨁⨁◯◯ LOW	OR 1.22 [0.19; 7.77]
PHLF	3 (2 RCTs, 1 retrospective)	⨁⨁◯◯ LOW	OR 0.77 [0.07; 8.86]
Bile leakage	6 (4 RCTs 2 retrospective)	⨁⨁⨁◯ Moderate	OR 1.31 [0.62; 2.78]

CI: Confidence interval; OR: odds ratio.

## Data Availability

All data generated or analyzed during this study are included in this published article and its Appendix A.

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
