# Peer review of "Routine Postoperative Antibiotic Prophylaxis Offers No Benefit after Hepatectomy—A Systematic Review and Meta-Analysis"

_antibiotics, 2022, doi:10.3390/antibiotics11050649_

Round 1
Reviewer 1 Report
There is a previous meta-analysis in the literature (Guo T, Ding R, Yang J, Wu P, Liu P, Liu Z, Li Z. Evaluation of different antibiotic prophylaxis strategies for hepatectomy: A network meta-analysis. Medicine (Baltimore). 2019 Jun; 98 (26): e16241. Doi: 10.1097 / MD.0000000000016241. PMID: 31261586; PMCID: PMC6617204.) Which analyzed only 4 randomized studies (2 have added in the meantime) and which already reached the same conclusions. This previous meta-analysis should be cited, emphasizing that the present meta-analysis is an update of the previous one and which consolidates the results of that one.

Author Response
Dear Editor and Reviewers,
We thank you for taking the time to review our manuscript. We value the comments provided and hope to have addressed them to your satisfaction. We feel that the manuscript has benefited from the revision. Please, find a point-by-point response to your comments below.
On behalf of all authors,
Dr. A. Murtha-Lemekhvoa
Reviewer: 1
We thank the reviewer for their time and overall positive response.
Comments:
There is a previous meta-analysis in the literature (Guo T, Ding R, Yang J, Wu P, Liu P, Liu Z, Li Z. Evaluation of different antibiotic prophylaxis strategies for hepatectomy: A network meta-analysis. Medicine (Baltimore). 2019 Jun; 98 (26): e16241. Doi: 10.1097 / MD.0000000000016241. PMID: 31261586; PMCID: PMC6617204.) Which analyzed only 4 randomized studies (2 have added in the meantime) and which already reached the same conclusions. This previous meta-analysis should be cited, emphasizing that the present meta-analysis is an update of the previous one and which consolidates the results of that one.
Answer:
We thank the reviewer for the remark. We have included the network meta-analysis into the discussion and emphasized that the current meta-analysis confirms the findings. Additionally, current meta-analysis suggests an increase in resistance of pathogens is associated with prolonged antibiotic prophylaxis.
Line 235-237 “Previous network meta-analysis of five studies has compared various antibiotic strategies and did not reveal a benefit of additional antibiotics, findings which were confirmed in the current meta-analysis of nine studies (27).”
Line 241-243 “Current meta-analysis suggests, increase of resistance in pathogens after postoperative antibiotic prophylaxis without benefit of reduced rate of overall infections.”
Reviewer 2 Report
Postoperative antibiotic prophylaxis offers no benefit after hepatectomy – a systematic review and meta-analysis
The title should avoid stating directly the conclusion of the study, especially when the results are not conclusive. It directly implies that post-op prophylactic usage of antibiotics is not an essential part to prevent infection in selective hepatectomy-which is not 100% confirmed
Abstract
Background Prophylactic antibiotics are frequently administered after major abdominal surgery including hepatectomies aiming to prevent infective complications. Yet, excessive use of antibiotics increases resistance in bacteria. The aim of this systematic review and meta-analysis is to assess the efficacy of prophylactic antibiotics after hepatectomy (POA). Method This systematic review and meta-analysis was completed according to the current PRISMA guidelines. The protocol has been registered prior to data extraction. MEDLINE, Web of Science and CENTRAL were searched for clinical reports on POA in hepatectomy restrictions. A random effects model was used for synthesis. Methodological quality was assessed with RoB2 and ROBINS-I. GRADE was used for quality of evidence assessment. Results Nine comparative studies comprising 2987 patients were identified: six RCTs and three retrospective. POA did not lead to a reduction in postoperative infective complications or have an effect on liver-specific complications – post-hepatectomy liver failure and biliary leaks. POA over four or more days was associated with increased rates of deep surgical site infections compared to short-term administration for up to two days (OR 1.54; 95%CI [1.17;2.03]; p=0.03). Postoperative antibiotic prophylaxis led to significantly higher MRSA incidence as pathogen (p=0.0073). Overall, the risk of bias in the studies was low and quality of evidence moderate. Conclusion Routine POA cannot be recommended after hepatectomy since it does not reduce postoperative infective or liver-specific complications but contributes to resistance in bacteria. Studies into individualized risk adapted antibiotic prophylaxis strategies are needed to further optimize perioperative treatment in liver surgery.
Comments: there are some misleading statements
Prophylactic antibiotics are frequently administered after major abdominal surgery including hepatectomies aiming to prevent infective complications-
There are four antibiotic prophylaxis strategies, namely, preoperative application (PRA, defined as short-duration administration of antibiotics prior to skin incision), postoperative short-duration application (POS, defined as ≤2 days postoperative application), postoperative long-duration application (POL, defined as >2 days postoperative application), negative control (NC, indicating no antibiotic prophylaxis) and their combination
-which one that you are referring too.
POA refers to postoperative antibiotics-this should be mentioned in the first instance
This meta analysis aimed to examine the evidence for the use of antimicrobial prophylaxis in liver surgery. It must be noted that previously an Cochrane meta-analysis including 7 RCTs showed that no antimicrobial method could improve outcomes after hepatectomy. Indeed, preoperative antibiotic prophylaxis is administered routinely in many liver surgery centers. However, most of these results were based on uncontrolled retrospective studies. A recent prospective RCT showed that prophylactic antibiotics resulted in no statistically significant benefit for total infections, surgical site infection and remote site infection. The findings may attribute to the progressively improved surgical technique and other non-antibiotic-based physical prophylactic procedures, as infectious complications are usually related to technical pitfalls rather than the use of prophylactic antibiotics.
A meta-analysis published in 2004 provided the strongest evidence to omit routine prophylactic drainage after major abdominal surgery, although only 3 RCTs on liver resection with low sample size were included in this meta-analysis. Furthermore, several retrospective cohort studies and randomized controlled trials have suggested that abdominal drainage after liver resection may increase the risk of complications such as wound infection, retrograde abdominal infection, and ascitic fluid leakage. However, many hepatic surgeons still continue to use routine drainage after hepatic resection for early detection of hemorrhage or bile leakage and reduce need for re-intervention in clinical practice.
I would like to recommend the author to discuss the gap of research and highlight the important point why this review is different from the predecessor.
Independent risk factors for infectious complications such as long operation time, blood transfusion and bile leakage should be discussed
Author Response
Dear Editor and Reviewer,
We thank you for taking the time to review our manuscript. We value the comments provided and hope to have addressed them to your satisfaction. We feel that the manuscript has benefited from the revision. Please, find a point-by-point response to your comments below.
On behalf of all authors,
Dr. A. Murtha-Lemekhova
Comments:
Postoperative antibiotic prophylaxis offers no benefit after hepatectomy – a systematic review and meta-analysis
The title should avoid stating directly the conclusion of the study, especially when the results are not conclusive. It directly implies that post-op prophylactic usage of antibiotics is not an essential part to prevent infection in selective hepatectomy-which is not 100% confirmed
Answer: We thank the reviewer for their comment. We have edited the title to improve clarity that routine postoperative antibiotics have not shown to be beneficial based on the results of this analysis.
Line 1-3 “Routine postoperative antibiotic prophylaxis offers no benefit after hepatectomy – a systematic review and meta-analysis”
Abstract
Background Prophylactic antibiotics are frequently administered after major abdominal surgery including hepatectomies aiming to prevent infective complications. Yet, excessive use of antibiotics increases resistance in bacteria. The aim of this systematic review and meta-analysis is to assess the efficacy of prophylactic antibiotics after hepatectomy (POA). Method This systematic review and meta-analysis was completed according to the current PRISMA guidelines. The protocol has been registered prior to data extraction. MEDLINE, Web of Science and CENTRAL were searched for clinical reports on POA in hepatectomy restrictions. A random effects model was used for synthesis. Methodological quality was assessed with RoB2 and ROBINS-I. GRADE was used for quality of evidence assessment. Results Nine comparative studies comprising 2987 patients were identified: six RCTs and three retrospective. POA did not lead to a reduction in postoperative infective complications or have an effect on liver-specific complications – post-hepatectomy liver failure and biliary leaks. POA over four or more days was associated with increased rates of deep surgical site infections compared to short-term administration for up to two days (OR 1.54; 95%CI [1.17;2.03]; p=0.03). Postoperative antibiotic prophylaxis led to significantly higher MRSA incidence as pathogen (p=0.0073). Overall, the risk of bias in the studies was low and quality of evidence moderate. Conclusion Routine POA cannot be recommended after hepatectomy since it does not reduce postoperative infective or liver-specific complications but contributes to resistance in bacteria. Studies into individualized risk adapted antibiotic prophylaxis strategies are needed to further optimize perioperative treatment in liver surgery.
Comments: there are some misleading statements
Prophylactic antibiotics are frequently administered after major abdominal surgery including hepatectomies aiming to prevent infective complications-
There are four antibiotic prophylaxis strategies, namely, preoperative application (PRA, defined as short-duration administration of antibiotics prior to skin incision), postoperative short-duration application (POS, defined as ≤2 days postoperative application), postoperative long-duration application (POL, defined as >2 days postoperative application), negative control (NC, indicating no antibiotic prophylaxis) and their combination
-which one that you are referring too.
Answer: We thank the reviewer for their comment. We agree that there are four antibiotic strategies, as stated by the reviewer. In the paragraph on the background to the study, we refer to antibiotic prophylaxis administered after hepatectomies, which comprise postoperative short-duration and postoperative long-duration antibiotics.
POA refers to postoperative antibiotics-this should be mentioned in the first instance
Answer: We thank the reviewer for their comment. We have added that POA refers to postoperative antibiotics.
Page 1: “The aim of this systematic review and meta-analysis is to assess the efficacy of prophylactic antibiotics after hepatectomy (postoperative antibiotic prophylaxis, POA).”
This meta analysis aimed to examine the evidence for the use of antimicrobial prophylaxis in liver surgery. It must be noted that previously an Cochrane meta-analysis including 7 RCTs showed that no antimicrobial method could improve outcomes after hepatectomy. Indeed, preoperative antibiotic prophylaxis is administered routinely in many liver surgery centers. However, most of these results were based on uncontrolled retrospective studies. A recent prospective RCT showed that prophylactic antibiotics resulted in no statistically significant benefit for total infections, surgical site infection and remote site infection. The findings may attribute to the progressively improved surgical technique and other non-antibiotic-based physical prophylactic procedures, as infectious complications are usually related to technical pitfalls rather than the use of prophylactic antibiotics.
Answer: We thank the reviewer for their comment. The Cochrane Review by Gurusamy et al (2011) analyzed various anti-infective strategies, namely antibiotics versus none, long duration versus short duration, prebiotics and probiotics versus none, preoperative and postoperative prebiotics and probiotics versus postoperative prebiotics and probiotics, topical versus no topical povidone iodine gel, and the included outcomes were mortality and serious adverse events (primary outcomes) and hospital stay (secondary outcomes). The Cochrane review did not investigate effect of these interventions on surgical site infections, total or remote infections, or resistance of bacteria. Additionally, the interventions compared were chosen differently, as we have compared perioperative antibiotic prophylaxis, which is recommended by WHO and is administered in liver surgery centers, to the additional postoperative prophylaxis, which lacks such recommendation. Current meta-analysis included seven more studies which have been published after the Cochrane Review, which included two RCTs – one comparing 7-day postoperative antibiotic prophylaxis versus none, and the other one comparing 2-day postoperative prophylaxis versus 5-day.
We also agree with the reviewer that improved surgical technique may contribute to the reduced surgical-site infections and discuss other factors which have previously been associated with SSIs. We believe that in addition to surgical technique, perioperative management should be further investigated.
Page 2: “Despite standards for hygiene, operations lasting longer than 3 hours, those associated with high blood loss and blood transfusions, or requiring a long period of anesthesia are a major risk factor for postoperative infections (2). Moreover, length of stay in the intensive care unit, as well as the total length of hospital stay, systemic comorbidities, previous antibiotic therapy and the degree of contamination of the operation are all risk factors for SSI (4).”
Page 6: “Liver surgery is a rapidly growing surgical specialty with expanding indications and novel techniques appearing and constantly evolving (21, 22). In addition to technique optimization, it is important to improve perioperative management. With complication rates reaching 31% for hepatectomy, validated strategies to reduce morbidity are wanted (23). Postoperative antibiotic prophylaxis has been a subject of debate recently as more and more studies failed to show benefit of this practice even in cases of intraoperative contamination (24-26).”
A meta-analysis published in 2004 provided the strongest evidence to omit routine prophylactic drainage after major abdominal surgery, although only 3 RCTs on liver resection with low sample size were included in this meta-analysis. Furthermore, several retrospective cohort studies and randomized controlled trials have suggested that abdominal drainage after liver resection may increase the risk of complications such as wound infection, retrograde abdominal infection, and ascitic fluid leakage. However, many hepatic surgeons still continue to use routine drainage after hepatic resection for early detection of hemorrhage or bile leakage and reduce need for re-intervention in clinical practice.
I would like to recommend the author to discuss the gap of research and highlight the important point why this review is different from the predecessor.
Answer: We thank the reviewer for their comment. We agree that prophylactic drainage after major abdominal surgery and in particular hepatectomies needs further investigation. This meta-analysis did not aim to investigate the effect of drainage on surgical site infections or other complications. Routine use of drainage was also not an additionally investigated factor in included studies. The reviewer may find a recent meta-analysis by Dezfouli et al interesting, as it investigated the association between wound infections and drainage in major hepatectomy and found no difference between two groups with and without an abdominal drain (Dezfouli SA, Ünal UK, Ghamarnejad O, et al. Systematic review and meta-analysis of the efficacy of prophylactic abdominal drainage in major liver resections. Sci Rep. 2021;11(1):3095. Published 2021 Feb 4. doi:10.1038/s41598-021-82333-x).
Independent risk factors for infectious complications such as long operation time, blood transfusion and bile leakage should be discussed
Answer:
We thank the reviewer for the remark. We discuss independent risk factors such as those mentioned by the reviewer in the introduction of our paper.
Line 45-50 “Despite standards for hygiene, operations lasting longer than 3 hours, those associated with high blood loss and blood transfusions, or requiring a long period of anesthesia are a major risk factor for postoperative infections (2). Moreover, length of stay in the intensive care unit, as well as the total length of hospital stay, systemic comorbidities, previous antibiotic therapy and the degree of contamination of the operation are all risk factors for SSI (4). “
Reviewer 3 Report
Reviewer comments
Murtha-Lemekhova and colleagues performed a meta-analysis of 9 studies with 2,987 patients, and suggested that the postoperative antibiotic prophylaxis (POA) has no benefit for hepatectomy. Interestingly, they showed that routine POA did not lead to a reduction in postoperative infective complications or have an effect on liver-specific complications – post-hepatectomy liver failure and biliary leaks. A long-term POA for 4~5 days had increased rates of deep surgical site infections compared to short-term administration for up to two days. Authors suggested that prolonged POA was associated with significantly higher MRSA incidence. However, only two studies compared the two regimens – POA2 and POA4 – to show that deep surgical site infections occurred significantly more frequently in patients receiving longer regimens. In these cases, however, authors needed to use the two studies in order to show the effect of prolonged POA on the increase of deep surgical site infection with MRSA. Though it is the major point that they take into consideration, most statistic results end up being too much descriptive to be able to draw out a solid conclusion on the effect of POA on hepatectomy. If a small number of studies conducted under similar conditions with similar subjects had low I², a fixed effects model might be appropriate to examine as a whole especially in Figure 3b, Figure 4a, Figure 4b, Figure 4c, and Table 5 with PHLF and sepsis downgraded one level (lines 225-226). The combinability of the two studies with the other studies appeared to be disputable and ill-defined. No matter whether it was fixed or random, authors might require generalization with additional analysis of unpredicted studies under conditions of high complexity, in order to support their finding (lines 250-251). What matters, most of the Discussion is not based on these analyses to interpret their findings being similar or different in the tested and other studies.
Specific comments:
In abstract
Line 24, RCTs -> randomized controlled trials
Line 28, Postoperative antibiotic prophylaxis -> It
Line 31, infective - > infection
In Introduction
Line 39-40, does this mean? “Once SSI occurs, the mortality drastically increases by over 12% and reducing it is a priority”
Line 51, situs -> site
Line 56, What require redosing? Specifically rewrite the vague sentence.
Line 62, 50% after hepatectomy -> “ 50% patients receiving hepatectomy
In Methods,
Line 109, input “and” before microbiota changes.
In Results
Line 212, italicize “Staphylococcus aureus”
Line 213, Staphylococcus aureus -> S. aureus
Line 215, staphylococcus aureus -> S. aureus
Line 225, remove “only”.
In Figure 6, what is the number of the y-axis? If it is a sum of pathogens identified in each study, it would rather show the plots of frequencies of all studies. Moreover, it is important to show the positive relationship between deep surgical site infection and MRSA.
Author Response
Dear Editor and Reviewer,
We thank you for taking the time to review our manuscript. We value the comments provided and hope to have addressed them to your satisfaction. We feel that the manuscript has benefited from the revision. Please, find a point-by-point response to your comments below.
On behalf of all authors,
Dr. A. Murtha-Lemekhova
Reviewer comments
Murtha-Lemekhova and colleagues performed a meta-analysis of 9 studies with 2,987 patients, and suggested that the postoperative antibiotic prophylaxis (POA) has no benefit for hepatectomy. Interestingly, they showed that routine POA did not lead to a reduction in postoperative infective complications or have an effect on liver-specific complications – post-hepatectomy liver failure and biliary leaks. A long-term POA for 4~5 days had increased rates of deep surgical site infections compared to short-term administration for up to two days. Authors suggested that prolonged POA was associated with significantly higher MRSA incidence. However, only two studies compared the two regimens – POA2 and POA4 – to show that deep surgical site infections occurred significantly more frequently in patients receiving longer regimens. In these cases, however, authors needed to use the two studies in order to show the effect of prolonged POA on the increase of deep surgical site infection with MRSA. Though it is the major point that they take into consideration, most statistic results end up being too much descriptive to be able to draw out a solid conclusion on the effect of POA on hepatectomy. If a small number of studies conducted under similar conditions with similar subjects had low I², a fixed effects model might be appropriate to examine as a whole especially in Figure 3b, Figure 4a, Figure 4b, Figure 4c, and Table 5 with PHLF and sepsis downgraded one level (lines 225-226). The combinability of the two studies with the other studies appeared to be disputable and ill-defined. No matter whether it was fixed or random, authors might require generalization with additional analysis of unpredicted studies under conditions of high complexity, in order to support their finding (lines 250-251). What matters, most of the Discussion is not based on these analyses to interpret their findings being similar or different in the tested and other studies.
Answer: We thank the reviewer for their comment and their expertise. There is an ongoing discussion about the utilization of fixed and random effects model in meta-analyses. While some authors opt for a fixed effects model in cases of low heterogeneity as estimated by I2, in cases of methodological heterogeneity in study design, interventions or outcome measures, a random-effects model should be utilized (Tufanaru, Catalin MD, MPH, MClinSci (EBHC); Munn, Zachary PhD; Stephenson, Matthew PhD; Aromataris, Edoardo PhD Fixed or random effects meta-analysis? Common methodological issues in systematic reviews of effectiveness, International Journal of Evidence-Based Healthcare: September 2015 - Volume 13 - Issue 3 - p 196-207 doi: 10.1097/XEB.0000000000000065). Additionally, as Tufanaru et al suggest, to generalize the results beyond included studies, a random-effects model needs to be applied. For these two reasons, we saw a random-effects model as the best applicable methodology.
The main result of the meta-analysis, based on seven studies, is that additional postoperative antibiotic prophylaxis does not offer benefit. An additional analysis of the two studies investigating prolonged postoperative antibiotic prophylaxis showed an association with more deep surgical site infections. The studies with prolonged postoperative antibiotic prophylaxis were not included into the main analysis, as this would harm the accuracy and precision of the results (s. Figure 2 and Figure 4), as was also pointed out by the reviewer. The data on isolated pathogens was provided by six studies investigating postoperative antibiotic prophylaxis versus control, the two studies with prolonged POA did not contribute to this section. We have included a clarification into the text.
Line 208-209 “Most common pathogenic isolates for control versus POA were provided by six studies comparing POA versus control (12, 14-16, 18, 20) and are depicted in Figure 6.”
We also thank the reviewer for their remark on discussion and revised it for improvement. In the discussion, we added a section on this meta-analysis confirming the results of a network meta-analysis by Guo et al. and added a section on resistance of pathogens.
Lines 235-237 “Previous network meta-analysis of five studies has compared various antibiotic strategies and did not reveal a benefit of additional antibiotics, findings which were confirmed in the current meta-analysis of nine studies (28)”
Lines 241-243 “Current meta-analysis suggests, increase of resistance in pathogens after postoperative antibiotic prophylaxis without benefit of reduced rate of overall infections.”
Specific comments:
In abstract
Line 24, RCTs -> randomized controlled trials
Answer: We thank the reviewer for their remark and have clarified it in the text.
Line 28, Postoperative antibiotic prophylaxis -> It
Answer: We thank the reviewer for their remark and have exchange “Postoperative antibiotic prophylaxis” for “POA”
Line 31, infective - > infection
Answer: We thank the reviewer for their remark and have exchanged the word “infective” for “infection”
In Introduction
Line 39-40, does this mean? “Once SSI occurs, the mortality drastically increases by over 12% and reducing it is a priority”
Answer: We thank the reviewer for their comment. Once SSI occurs, the mortality increases by over 12%, thus strategies to reduce SSIs or prevent them is a priority. To increase clarity, we have changed the sentence as follows: In patients with SSI, mortality increases by over 12%, making strategies to prevent SSI a priority (1).
Line 51, situs -> site
Answer: We thank the reviewer for their comment. We have changed the word accordingly.
Line 56, What require redosing? Specifically rewrite the vague sentence.
Answer: We thank the reviewer for their comment and have re-written the sentence to clarify that redosing refers to perioperative antibiotics. “This instigates questions whether abdominal surgeries with lower contamination potential require redosing of perioperative antibiotics at all.”
Line 62, 50% after hepatectomy -> “ 50% patients receiving hepatectomy
Answer: We thank the reviewer for their comment. We have changed the sentence accordingly
In Methods,
Line 109, input “and” before microbiota changes.
Answer: We thank the reviewer for their comment. We have changed the sentence accordingly
In Results
Line 212, italicize “Staphylococcus aureus”
Answer: We thank the reviewer for their comment. We have changed the words accordingly
Line 213, Staphylococcus aureus -> S. aureus
Answer: We thank the reviewer for their comment. We have changed the words accordingly
Line 215, staphylococcus aureus -> S. aureus
Answer: We thank the reviewer for their comment. We have changed the words accordingly
Line 225, remove “only”.
Answer: We thank the reviewer for their comment. We have changed the sentence accordingly
In Figure 6, what is the number of the y-axis? If it is a sum of pathogens identified in each study, it would rather show the plots of frequencies of all studies. Moreover, it is important to show the positive relationship between deep surgical site infection and MRSA.
Answer: We thank the reviewer for their remark. We have updated the figure to clarify that the y-axis depicts number of pathogenic isolates. These findings were pooled from the included studies to analyze the association between POA and bacterial resistance. As authors from the individual included studies have not provided raw data, a positive relationship between deep SSIs and MRSA cannot tested at this point.
Round 2
Reviewer 3 Report
Reviewer comments:
Murtha-Lemekhova and colleagues revised the manuscript (antibiotics-1708031). They suggested that a random-effects model should be utilized in cases of methodological heterogeneity in study design. For the previous comment on Figure 6, authors showed that the y-axis depicts number of pathogenic isolates. Still, I cannot understand well the major point that why and how they pooled the numbers of bacterial isolates from the included studies, if the individual studies have not provided raw data. Because authors modified the title, they need to modify the sentences “Routine POA led to significantly higher MRSA incidence as pathogen (p=0.0073)” in Abstract (lines 28-29), “with routine POA, while S. aureus (without resistance) was …” in Results (lines 214-215). Because there were bias between the included studies, authors could add their comment “The individual studies were not sufficient to test a positive relationship between deep SSIs and MRSA.”
Specific comments:
Lines 255-256, does it mean? “the long-term treatment of antibiotics may not be effective and can cause the selection of resistant bacteria including MRSA.” Remove “rather than shield the organism from it.” Remove also “ Selecting bacteria … leads to increased multi-drug resistance in Staphylococcus aureus” in lines 256-262, because it is unclear that antibiotics lead to an increase of (multi-drug) resistance in bacteria, rather than the selection of them.
Lines 263-264, according to the above comment, the sentence can be modified as “Long-term consequences of increased MRSA in patients after hepatectomy, too, remains unclear …”

Author Response
Response to reviewers
Dear Editor and Reviewer,
We thank you for taking the time to review our manuscript. We value the comments provided and hope to have addressed them to your satisfaction. We feel that the manuscript has benefited from the revision. Please, find a point-by-point response to your comments below.
On behalf of all authors,
Dr. A. Murtha-Lemekhova
Reviewer comments
Murtha-Lemekhova and colleagues revised the manuscript (antibiotics-1708031). They suggested that a random-effects model should be utilized in cases of methodological heterogeneity in study design. For the previous comment on Figure 6, authors showed that the y-axis depicts number of pathogenic isolates. Still, I cannot understand well the major point that why and how they pooled the numbers of bacterial isolates from the included studies, if the individual studies have not provided raw data.
Answer: We thank the reviewer for their remark and are happy to clarify. Individual studies provided distribution of pathogens responsible for SSIs, however not the raw data concerning the location where the isolate has been extracted from (superficial, deep SSI).Therefore an association of MRSA with deep SSI cannot be tested for.
Because authors modified the title, they need to modify the sentences “Routine POA led to significantly higher MRSA incidence as pathogen (p=0.0073)” in Abstract (lines 28-29), “with routine POA, while S. aureus (without resistance) was …” in Results (lines 214-215).
Answer: We thank the reviewer for the remark and have modified the sentences accordingly.
Because there were bias between the included studies, authors could add their comment “The individual studies were not sufficient to test a positive relationship between deep SSIs and MRSA.”
Answer: We thank the reviewer for this comment and have added a sentence in the results section. Lines 219-220: The association between deep SSIs and MRSA could not be tested due to insufficient raw data provided by individual studies.
Specific comments:
Lines 255-256, does it mean? “the long-term treatment of antibiotics may not be effective and can cause the selection of resistant bacteria including MRSA.” Remove “rather than shield the organism from it.” Remove also “ Selecting bacteria … leads to increased multi-drug resistance in Staphylococcus aureus” in lines 256-262, because it is unclear that antibiotics lead to an increase of (multi-drug) resistance in bacteria, rather than the selection of them.
Answer: We thank the reviewer for their comment. We have removed “rather than shield the organism from it and modified the sentence. Concerning increased multi-drug resistance, the article cited at that point provides an overview how selection of bacteria leads to emergence of multidrug-resistant strains. In short, Medina et al. describe how under antibiotic selection pressure, bacteria develop or acquire genes that provide resistance to multiple antibiotics.
Lines 263-264, according to the above comment, the sentence can be modified as “Long-term consequences of increased MRSA in patients after hepatectomy, too, remains unclear …”
Answer: Thank you for your comment. We have modified the sentence to emphasize MRSA. “Long-term consequences of increased resistant bacteria, in particular MRSA, in patients after hepatectomy, too, remains unclear…”